# Comparison of the Portuguese Version of the Pregnancy Physical Activity Questionnaire (PPAQ) with Accelerometry for Classifying Physical Activity among Pregnant Women with Obesity

**DOI:** 10.3390/ijerph20020929

**Published:** 2023-01-04

**Authors:** Diana Bernardo, Carlos Carvalho, Raquel Leirós-Rodríguez, Jorge Mota, Paula Clara Santos

**Affiliations:** 1KinesioLab Research Unit in Human Movement, Department of Physiotherapy, Piaget Institute, School of Health, 4405-678 Vila Nova de Gaia, Portugal; 2Research Centre in Physical Activity, Health and Leisure (CIAFEL), Faculty of Sport, University of Porto (FADEUP), 4200-450 Porto, Portugal; 3Sword Health Technologies, Department of Physiotherapy, 4100-467 Porto, Portugal; 4SALBIS Research Group, Nursing and Physical Therapy Department, University of Leon, 24004 León, Spain; 5Laboratory for Integrative and Translational Research in Population Health (ITR), Faculty of Sport, University of Porto (FADEUP), 4200-450 Porto, Portugal; 6Department of Physiotherapy, School of Health, Polytechnic of Porto (ESS), 4200-072 Porto, Portugal; 7Center for Rehabilitation Research (CIR), School of Health, Polytechnic of Porto, 4200-072 Porto, Portugal

**Keywords:** pregnancy, maternal, obesity, assessment, physical activity, gynecology and obstetrics

## Abstract

In recent years, the number of pregnant women with obesity has increased exponentially; thus, it is important to evaluate and characterize the physical activity levels of this specific group. The aim of this study is to evaluate the reliability and validity of the Portuguese version of the Physical Activity and Pregnancy Questionnaire and Pregnancy Questionnaire in pregnant women with obesity and to classify physical activity using the Physical Activity and Pregnancy Questionnaire and accelerometry. An analytical observational study was carried out between May and August of 2019 at the University Hospital Center of São João, with a sample of 31 pregnant women with obesity (30.9 ± 4.6 years 36.5 ± 4.6 kg/m^2^ of BMI and 21.5 ± 9 gestational weeks). The physical activity of participants was evaluated using an accelerometer and Physical Activity and Pregnancy Questionnaire at two time points (the first visit at the moment of consultation and the second seven days after, with accelerometer retest), the interclass correlation coefficient was used to test reliability between the Physical Activity and Pregnancy Questionnaire filled out at visit1 and the Physical Activity and Pregnancy Questionnaire filled out at visit2, and Pearson’s correlation was used to determine validity between the Physical Activity and Pregnancy Questionnaire and accelerometry. The interclass correlation coefficient values for total activity were 0.95, 0.97 for moderate and 0.58 for vigorous intensities. It ranged from 0.74 for sports/exercise to 0.96 for domestic activities. The Pearson’s correlations showed that the Physical Activity and Pregnancy Questionnaire is moderately valid for moderate intensity (r = 0.435). A total of 67.7% of the pregnant women complied with international physical activity recommendations.

## 1. Introduction

In recent years, industrialized nations have experienced an exponential increase of obesity, including in women of reproductive age [1,2]. It is estimated that the prevalence of maternal obesity is between 8 and 30% [1,3,4], and that 50% of women become pregnant with a Body Mass Index (BMI) that can be categorized as pre-obesity [5,6].

Habits adopted by women during pregnancy can affect their health for the rest of their lives [7]. As a result, pregnancy, which is associated with profound anatomical and physiological changes, is also recognized as a unique moment for behavior modification and is not in itself considered a physical condition requiring restriction of physical activity (PA) practice [8]. In fact, the Guidelines for Gynecology and Obstetrics recommend the practice of PA during pregnancy and the incorporation of its practice by sedentary women prior to pregnancy [9].

PA plays a key role in health, and during pregnancy its practice has minimal risk [10], and it is closely associated with reducing the risk of maternal complications (like gestational diabetes, preeclampsia, postpartum depression, etc.) [10,11]. Therefore, the current recommendation is that all women without contraindications should be physically active throughout pregnancy (and subgroups of women with contraindications should be appropriately screened and counseled in a personalized manner). In general, pregnant women should accumulate at least 150 min of moderate-intensity PA each week to achieve clinically significant health benefits and reduce pregnancy complications. In addition, PA should be accumulated for a minimum of 3 days per week; however, activity every day is recommended [12].

One study conducted in Portugal shoed that Portuguese pregnant women did not have sports habits throughout pregnancy, and that PA decreases with progress of pregnancy [13]. Other studies have revealed that high pre-gestational BMI and maternal weight gain influence neonatal outcomes [14], and are associated with an increased risk of overweight/obesity in childhood with noticeable effects at later ages [15]. Given the exponential worldwide increase in maternal obesity [4,16,17], and since pregnant women with obesity spend more time in light-intensity activities, but see it as being moderate or vigorous [18], it is important to subjectively and objectively evaluate and characterize the PA levels of this specific group in order to plan interventions tailored to their needs. 

Proper assessments of PA are important for determining both trends and health benefits and their effects over time [19]. PA can be evaluated using objective methods, using pedometers and accelerometers and/or subjectively on the basis of questionnaires [20]. Moreover, within these two different assessment modalities, there is great heterogeneity in the way assessment instruments are applied [21].

However, there is a scarcity of PA measurement instruments in pregnant women with obesity [18]. The Pregnancy Physical Activity Questionnaire (PPAQ) aims to measure the duration, frequency and intensity of domestic, occupational, sports and transportation activities, and provides a quantitative measure of types of PA intensity, including sedentary lifestyle [22]. The questionnaire has been translated and adapted into several languages, and demonstrates excellent reliability. However, only the French validation focused on the evaluation of pregnant women with obesity [18].

Therefore, the objectives of this study are (a) to classify the PA of pregnant women with obesity using the PPAQ questionnaire and accelerometry, and (b) to evaluate the reliability and validity of the Portuguese version of the PPAQ in pregnant women with obesity.

## 2. Materials and Methods

### 2.1. Study Design and Participants

The applied research methodology consisted of an analytical and cross-sectional observational study to best describe the duration, frequency, and intensity of the total PA throughout pregnancy using the PPAQ. The study was carried out in accordance with the guidelines set out in the Declaration of Helsinki, and all women involved in the study signed a consent form prior to participation. The study was approved by the Ethics Committee of University Hospital Center of São João (code: 165/19). 

Inclusion criteria for participation in the research consisted of: (a) pregnant women between the 10th and 40th gestational week; (b) presenting a pre-gestational BMI equal to or greater than 30 kg/m^2^ (classified as obesity) [23]; and (c) attending the Gynecology and Obstetrics Department of the University Hospital Center of São João. The exclusion criteria were: (a) pregnant women under 18 years of age and over 40; (b) presenting previous bariatric surgery and/or neuromuscular diseases; (c) diagnosis of insulin-dependent diabetes, hypertension and/or cardiac diseases; (d) multiple pregnancy; and (e) women unable to follow or understand the procedure or who did not provide informed consent to participate in research [18,24]. The recruitment period took place between May and August of 2019.

### 2.2. Assessment Tools

(a) PPAQ: This questionnaire lists 32 activities; categories are organized into four intensity levels according to the score obtained in each and measured in metabolic equivalent (MET). One MET corresponds to the metabolic equivalent of energy expended at rest [22]. Sedentary activities correspond to energy expenditure < 1.5 METs; light activities between ≥1.5 and <3.0 METs; moderate activities between ≥3.0 and <6.0 METs; and vigorous energy expenditure ≥ 6.0 METs [22]. The energy expenditure on the activity in METs (intensity) is multiplied by duration of activity per day and thus obtains average measurement of energy spent weekly (MET-h·wk^−1^) [22]. 

(b) Accelerometers: The use of an accelerometer to assess PA levels is the gold standard with regard to the objectivity and reliability of measurement instruments [25]. The Actigraph GT3X^®^ and Actigraph wGT3X-BT^®^ accelerometer models (actigraphic, Pensacola, FL, USA) collect and analyze movements on three different axes: vertical, horizontal right–left and horizontal front and back [26]. According to Santos-Lozano et al. (2013), these models have been demonstrated to be an excellent tool for predicting expended energy levels in young people and adults, are considered appropriate to use during pregnancy, and have been used previously to evaluate PA in pregnant women with obesity [27].

### 2.3. Procedure

During the first visit, pregnant women filled in the sociodemographic questionnaire and the PPAQ (PPAQ_1_). Following that visit, the women received a portable accelerometer (Actigraph GT3X^®^ or Actigraph wGT3X-BT^®^ model (Actigraphic, Pensacola, FL, USA)) and were instructed to wear the device continuously for seven days. At the second visit, the same trained research assistant collected data from the accelerometer records and participants repeated the filling in of the PPAQ (PPAQ_2_).

The accelerometers were anonymized with an identification number and placed on the right antero-superior iliac crest [20,28]. The pregnant women were instructed to use it for 7 consecutive days [28,29] during waking hours (taken only for bathing or swimming, for sleeping or for discomfort), with only those who recorded at least 480 min of daily use and with the registration of two working days and at least one day of weekend or day off being considered valid [28,30]. The research team sent a daily SMS to encourage the use of the device. A daily log table was provided, where participants marked the time of accelerometer placement and removal as well as the activities carried out, which they assumed were underestimated by the accelerometer (e.g., static bike, weightlifting, swimming). The daily average of counts was considered only on days that participants met at least 480 min of use [18].

After seven days of use, the accelerometers were collected, and data were downloaded. To interpret the accelerometer data, we used Matthews’ cut-point protocol [31] (<759 counts/min—Light intensity (<3.0 METs); between 760–5725 counts/min—Moderate intensity (≥3.0 and <6.0 METs); ≥5726 counts/min—Vigorous intensity (≥6.0 METs)). This protocol was obtained through the combination of laboratory data with fieldwork on walks, races and the activities of daily living [31], and has been shown to be the most effective method to estimate energy expended [32]. This protocol was also used in another study with pregnant women with obesity [18].

Pre-gestational weight value was collected from the Pregnant Health Bulletin on the first visit and height obtained through a fixed stadiometer. Pregnant women with obesity were categorized according to World Health Organization into class I (BMI 30–34.9 kg/m^2^), class II (BMI between 35–34.9 kg/m^2^) and class III (BMI ≥ 40 kg/m^2^) [23].

### 2.4. Statistical Analysis

The Statistical Package for the Social Sciences (IBM SPSS Statistics^®^) version 20.0 (IBM Corporation, Armonk, NY, USA) program was used for statistical analysis. The analysis involved descriptive statistical measures (absolute and relative frequencies, means and their standard deviations) and inferential statistics. The significance level to reject the null hypothesis was set at (α) ≤ 0.05. The normal distribution of variables in the samples was accepted after the Shapiro–Wilk test, so parametric tests were used [33]. The Intraclass Correlation Coefficient and the Standard Measurement Error were used to test the reliability of the PPAQ. The values < 0.5 as “poor”, ≥0.5 and <0.75 as “moderate”, ≥0.75 and <0.9 as “good” and ≥0.90 with “excellent” [34] were considered [35]. Pearson’s coefficient (r) was used to test the validity between the PPAQ and the different accelerometer protocols, and the values of |r| between 0–0.3 as “weak”, |r| ≥ 0.3 and <0.6 as “moderate”, |r| ≥ 0.6 and <0.9 as “strong” and |r| ≥ 0.9 as “very strong” [35]. On PPAQ, the intensity below 3 METs is subdivided in light and sedentary. To correlate the PPAQ below 3 METs intensity with light intensity measured by accelerometry (also <3.0 METs), we used the sum of light and sedentary PPAQ intensities.

## 3. Results

### 3.1. Study Population

The initial sample consisted of 48 pregnant women. Sixteen of them did not accomplish the minimum of 480 min/day of accelerometer use, and one woman had a premature delivery. The final sample consisted of 31 pregnant women with a mean age of 30.9 ± 4.6 years (Table 1). Sociodemographic analysis found that the mean pre-pregnancy BMI was 36.5 kg/m^2^, and most of the pregnant women (67.7%) had a BMI equal to or higher than 35 kg/m^2^.

The mean gestational age of the pregnant women was 21.5 weeks; 13 (41.9%) pregnant women were in the second trimester and 10 (32.3%) in the third trimester. It was also found that 64.5% of participants were primiparous.

### 3.2. Physical Activity Reported by PPAQ

Table 2 presents the results of PPAQ1. The PA levels of pregnant women with obesity showed a means of 245.8 MET-h·wk^−1^ of total energy expended, with 59.8% of the activities being sedentary or light and 38.9% being related to domestic activities.

Moderate-intensity activities correspond to 39.8% of the total activities, and vigorous-intensity activities represent only 0.4% (1.29 MET-h·wk^−1^) of energy expenditure; regarding the type of activity, Sport/Exercise represents 2.2% (6.43 MET-h·wk^−1^) of energy expenditure. Domestic activity and moderate-intensity activity were positively skewed, and we found outliers particularly in the case of vigorous intensity (Figure 1).

### 3.3. Physical Activity Assessed by Accelerometry

Pregnant women walked an average of 5479.7 steps per day, and the accelerometer was used for 678.9 min per day on average (Table 3). Regarding time spent at different intensities, it was found that pregnant women spent 78.8% of their time (521.4 min·day^−1^) in light activities (<3 METs). Mean time spent in moderate activities was 152.2 min·day^−1^ (22.4%). The values of time spent on vigorous intensity were the lowest, not exceeding 0.8%. According to these data, 67.7% of the participants comply with the international recommendations of 150 min·wk^−1^.

### 3.4. PPAQ Rehability in Pregnant Women with Obesity

The reliability of the questionnaire was assessed using the intraclass correlation coefficient represented in Table 4. For total PPAQ activity, the intraclass correlation coefficient was 0.95, which is considered excellent [34]. For all other categories, except for light and vigorous intensity and sport/exercise activity, the values of the intraclass correlation coefficient were higher than 0.90. The vigorous intensity score had the lower intraclass correlation coefficient (0.58), which can be considered to indicate moderate reliability [34].

Agreement between PPAQ1 and PPAQ2 was illustrated using Bland–Altman plots (Appendix A). The limits of the agreement were calculated as the mean difference between the investigated indices ± 1.96 for standard deviation of difference.

### 3.5. PPAQ Validation through Comparison with Accelerometer

To test the validity of the PPAQ, the values of the PPAQ1 were compared with accelerometry (Table 5). The Pearson’s correlation coefficients of PPAQ1 for moderate activities were positive, moderate, and statistically significant (r = 0.435; *p* = 0.014).

## 4. Discussion

This study aimed to assess the validity and feasibility of PPAQ questionnaire and to classify PA in pregnant women with obesity using objective data, measured with accelerometers, and subjective data obtained through the PPAQ. 

It was found that the questionnaire was reliable for this specific population. The ICC, which was higher than 0.90 for most activities and intensities, represents excellent reliability [34]. The same results were found in the questionnaire adapted for the Turkish and French populations [18,36]. The lowest reliability values were obtained for the exercise/sport activity type and vigorous intensity (with low standard measurement error values). Results similar to these were obtained in PPAQ validation for the Chinese population. This may be due to the fact that the questionnaire is more sensitive when assessing lower PA intensity [37].

The mean values for the different types of activity are similar to those found in the Portuguese PPAQ validation [38], but slightly higher than the study by Chandonnet et al. (2012) [18], which aimed to measure the PA of pregnant women with obesity, mainly in terms of total and occupational activity. The fact that only 41% of the participants in the Chandonnet et al. (2012) [18] study were employed in the last trimester may have contributed to the low values of energy spent on occupational activity. It was also found that most activities of pregnant women with obesity were domestic and occupational, data which has been corroborated by other authors [13,39]. Regarding the accelerometer, the results observed in the current study for walked steps per day and time spent in activities below 3 METs are in line with those reported in other studies conducted with pregnant women [18,29,37]. Sedentary behavior is associated with non-favorable self-rated health during pregnancy [40], as well as high prenatal systolic and/or diastolic blood pressure [41], inadequate gestational weight gain [42], and shorter gestation and inhibited fetal growth [43]. In the general population, another study showed a positive relationship between low PA levels and hypokinetic and cardio-metabolic diseases [44]. Vigorous activity levels are extremely low in this specific population of pregnant women [45], and in women with obesity in general [46].

International guidelines for PA practice during pregnancy recommend that all pregnant women without medical contraindication should perform at least 150 min of moderate-intensity activity per week [12,47,48]; these recommendations were met by 67.7% of the pregnant women. These values are lower than those observed during the French PPAQ validation study for pregnant women with obesity [18] and in the Pebley et al. (2022) [49] study, where it was reported that 100% and 99% of participants, respectively, complied with the recommendations of PA practice. This may be because the sample in the present study used the accelerometer for an average of 678 min per day (approximately 11 h), while participants in the other studies [18,49] used accelerometers for 24 h, thus promoting higher values of time used. 

Nevertheless, the proportion of women who complied with international recommendations [12,47,48] was higher than anticipated. It is possible that the participants consciously or unconsciously increased their PA once they knew that they were being evaluated [20].

The correlation between the PPAQ and accelerometry was moderate (r = 0.435) for moderate intensity; however, the values for total activity can be considered weak (r = 0.271), with similar results being found during the validation of the original PPAQ [22]. One potential explanation for the weak values of correlation, aside from the recall bias of the questionnaire arising from the fact that PPAQ data refer to the last three months, may be the fact that the cut-off points for moderate and vigorous activity are based on non-pregnant and non-obese samples, and different cut-offs may be needed to adequately assess activity during pregnancy, given the changes experienced over the course of gestation [22,49].

The fact that 35.4% of eligible women were excluded for not having used the accelerometer may lead to a participation bias; however, we found homogeneity between these two groups regarding the BMI and age variables.

The results indicated that the PPAQ and accelerometers reported comparable PA levels for moderate intensity. However, future research with specific cut-point protocols for pregnant individuals with different BMI is needed.

## 5. Conclusions

The Portuguese version of the PPAQ questionnaire is reliable for the population of pregnant women with obesity and is moderately valid. For research requiring a detailed assessment of PA, the questionnaire can be used instead of accelerometers, since it is accurate, easy to apply, less expensive, and more accessible. 

Information obtained using the questionnaire might be helpful in monitoring health behaviors, as well as designing and promoting physical activity programs for pregnant women with obesity with the aim of potentiating a healthy lifestyle for both the mother and baby and preventing pregnancy complications.

## Figures and Tables

**Figure 1 ijerph-20-00929-f001:**
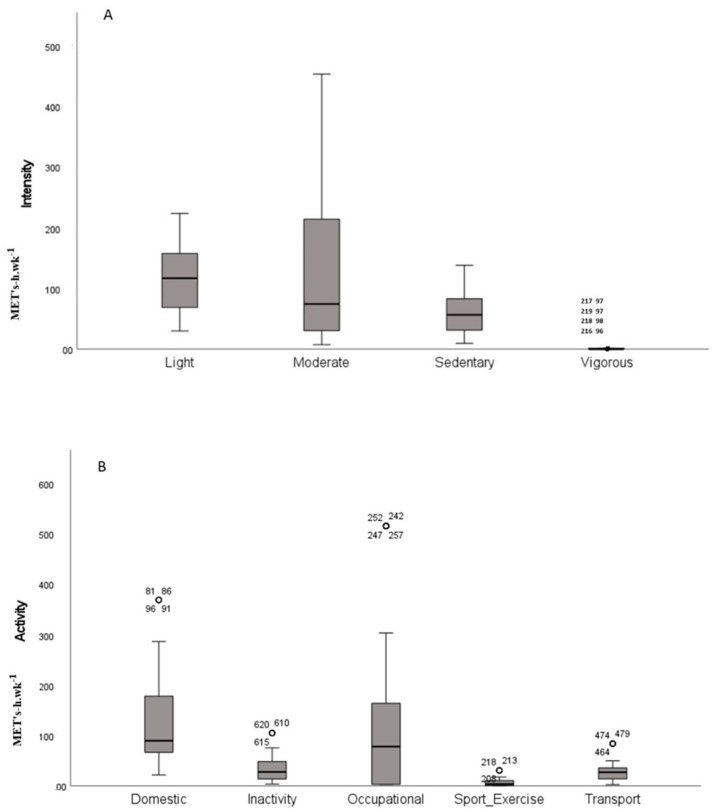
Dispersion results by type of intensity (**A**) and type of activity (**B**). The box plots illustrate the types of intensity (light, moderate, sedentary and vigorous) and the types of activity (domestic, inactivity, occupational, Sport/ Exercise and transport). The *y*-axes indicate the energy expended in METs.h·wk^−1^; the x-axis represents (**A**) the types of intensity and (**B**) the types of activity.

**Table 1 ijerph-20-00929-t001:** Sample characterization for sociodemographic and obstetric variables (data provided: mean ± standard deviation).

Variable	Mean ± Standard Deviation	n (Percentage)
Age (years)	30.9 ± 4.6	
Pre-pregnancy Body Mass Index (kg/m^2^)	36.5 ± 4.6	
Gestational age (weeks)	21.5 ± 9	
Pregnancy (number)	1.5 ± 2.9	
Marital status:		
Married/living with a partner		26 (83.9)
Single		5 (16.1)
Education:		
High school or less		19 (61.3)
College/graduate		12 (38.7)
Employment status:		
Employed during last trimester		26 (83.9)
Unemployed during last trimester		5 (16.1)
Economic monthly income:		
≤1250 €		20 (64.5)
≥1251 €		11 (35.5)

**Table 2 ijerph-20-00929-t002:** Average of the scores of the intensity and type of PA (METs.h·wk^−1^) self-reported through the PPAQ.

Intensity Type (METs.h·wk^−1^)				
	Mean ± Standard Deviation	Median	Percentiles 25–75%	Percentage
Total Intensity	295.6 ± 152	245.8	169.9–432.2	100
Sedentary	60.5 ± 35.5	56.7	31.5–83.3	20.5
Light	116.1 ± 53.6	117.3	69–158	39.3
Moderate	117.8 ± 121.1	73.3	30.7–214.7	39.8
Vigorous	1.3 ± 2.4	0	0–1.6	0.4
Activity Type (METs.h·wk^−1^)				
Domestic	117 ± 85.2	82.8	54.8–165.2	39.6
Occupational	105.2 ± 138.8	60	0–167.3	35.6
Sport/Exercise	6.4 ± 6.5	3.6	0.8–10.7	2.2
Transport	30.7 ± 16.4	28	19.3–35	10.4
Inactivity	36.2 ± 20.8	31.8	21.3–51	12.2

**Table 3 ijerph-20-00929-t003:** Steps, accelerometer usage time and physical activity intensities measured by accelerometer.

Variable	Mean ± Standard Deviation	Percentage
Steps (Steps·day^−1^)	5479.7 ± 2520.2	100
Total Activity (min/day)	678.9 ± 92.2	100
PA Intensity (min/day)		
Light	521.4 ± 65.1	76.8
Moderate	152.2 ± 62.8	22.4
Vigorous	5.3 ± 3.3	0.8

**Table 4 ijerph-20-00929-t004:** Reliability of the PPAQ—intraclass correlation coefficient (ICC) and the standard measurement error (SEM) between PPAQ1 and PPAQ2 (n = 31).

	ICC (95% IC)	SEM
**Total Activity (METs.h·wk^−1^)**	0.95 (0.9–0.98)	27.29
**Intensity (METs.h·wk^−1^)**		
● Sedentary	0.95 (0.89–0.98)	6.37
● Light	0.85 (0.69–0.92)	9.62
● Moderate	0.97 (0.93–0.98)	21.75
● Vigorous	0.58 (0.13–0.79)	0.45
**Type (METs.h·wk^−1^)**		
● Domestic	0.96 (0.92–0.98)	15.30
● Occupational	0.95 (0.89–0.98)	24.94
● Sport/Exercise	0.74 (0.47–0.88)	1.16
● Transport	0.82 (0.62–0.91)	3.40
● Inactivity	0.93(0.86–0.97)	4.42

ICC—intraclass correlation coefficient; IC—confidence index SEM—standard error measure; MET-h·wk ^−1^—unit of average energy expenditure per week.

**Table 5 ijerph-20-00929-t005:** Pearson’s correlation between PPAQ and accelerometer.

AF Intensity—PPAQ1	
Total	0.271
Sedentary + Light	0.096
Moderate	0.435 *
Vigorous	0.261

* *p* < 0.05.

## Data Availability

Not applicable.

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
