# Peer review of "Comparison of the Portuguese Version of the Pregnancy Physical Activity Questionnaire (PPAQ) with Accelerometry for Classifying Physical Activity among Pregnant Women with Obesity"

_ijerph, 2023, doi:10.3390/ijerph20020929_

Round 1

Reviewer 1 Report

The article entitled “Portuguese Version of the Pregnancy Physical Activity Questionnaire (PPAQ) compared with Accelerometry to Classify Physical Activity among Pregnant Women with obesity” aimed to (1) evaluate the reliability and validity of the Portuguese version of a Physical Activity (PA) Questionnaire on PA and Pregnancy (PPAQ) in pregnant women with obesity, and (2) classify the PA of obese pregnant women through the PPAQ questionnaire and accelerometry. The study concluded that (1) the PPAQ is reliable for obese pregnant women, and (2) one-third of the study participants did not comply with the ACSM recommendations for PA. Although the study is interesting, I have several concerns as follows, and some of them are major:

1.     The number of study participants is thirty-one (31), which is a very low number for attaining adequate statistical power. The authors did not mention any rationale in support of this.

2.     In line 121: It states, “Pregnant women with obesity were categorized according to Institute of Medicine into grade I (BMI ≥30; <35 Kg/m2), grade II (BMI ≥35; <39 Kg/m2) and grade III (BMI ≥ 40 Kg/m2) [28].” However, in the reference article [28], table 2-2 titled “Comparison of Institute of Medicine (IOM) and World Health Organization (WHO) BMI Categories” shows that IOM has only Obese Class I (>29 kg/m2), while WHO has Obese Class I, II and III. In addition, the values of these three classes do not match the “grades” mentioned by the present authors in the manuscript. Thus it is not clear on what basis the authors made the referenced statement.

3.     The quality/resolution of Fig. 1 is too low to be properly understood. Also, the y-axis labels are not properly typed and are occluded by the axes titles.

4.     Table 3 states three levels of PA but Table 4 shows four levels of activities. This is not clear in the text.

5.     In line 225: The authors mentioned “The mean values for the different types of activity are similar to those found in Portuguese validation, but slightly higher compared to the French study, mainly for total and occupational activity, the fact that only 41% of the participants of the French study were employed in the last trimester may have contributed to low energy values spent on occupational activity.” However, there is no reference stated and hence it is not clear which study has been referred to here.

6.     The participants in the study used the accelerometer for 505 minutes per day (~ 8 h). So, there is no data about the activity of the remaining 16 h of the day, which is a severe limitation of the study. This might also have confounded the findings and interpretation of the results in the present study.

7.     In line 253: The authors state “We can consider the questionnaire itself as a limitation to the present study, since the activities are classified through compendium and these values were obtained in a laboratory with adults (non-pregnant) [18].” This is another severe limitation of the study and makes the basic foundation of the study questionable. In other words, if the questionnaire itself is not accurate, how can the whole study be based on this?

8.     How is the domestic and occupational activity been classified and measured? Also, as pointed out above, the accelerometer was used only for ~ 8 h a day with no data available for the remaining 16 h of the day.

9.     The manuscript contains plenty of spelling mistakes, typographical and grammatical errors, incorrect sentence constructions, and incorrecter use of punctuations. It also contains sudden and abrupt changes of paragraphs, and in many places only one sentence is written in a paragraph format. Overall, it is very poorly written and needs extensive corrections. [For reference, one such example of an incorrect sentence construction is “After seven days of use, the accelerometers were collected and downloaded the data.”]

10.  In line 83: It states, “international PA recommendations”. The name of the article or the source is not mentioned. Assuming it is the ACSM, it needs to be specified. In addition, ACSM has not been spelled out at the first place in the article.

11.  The article also did not explain clearly about the energy expenditure calculation.

12.  The results are not written specifically. For example, the authors mentioned at one place “It was also found that most of the participants were primiparous.” without specifying the value. In other words, it is not clarified what is meant by “most” in the result, however, table 1 states it is 64.5%.

13.  The results did not specify gravida (i.e. the number of pregnancies), but only the parity (i.e. the number of live birth) of the pregnant participants.

14.  The table 1 shows “ = “ instead of “ ) “ in the Parity >= 1 row.

15.  In line 154: It states, “Moderate activities correspond to 39.9% of the total activities, the vigorous and sports activities were the ones with the lowest values, 0.4% (1.29 MET-h•wk-1) and 2.2% (6.43 MET-h•wk-1) respectively”. However, it is confusing why the authors mentioned “lowest” when two values (0.4% and 2.2%) are indicated.

16.  In Table 3: It shows the accelerometer use was 505.1 min per day, but the total activity was 678.1 min per day. It is not clear how the total activity was measured when the accelerometer was not used during the day.

17.  In line 193: It states, “The vigorous intensity score was the one with a lower Intraclass Correlation Coefficient of 0.58 considered moderate reliability”. Apart from the fact that the sentence is not constructed correctly, it is not clear how the value of 0.58 was considered. There is no reference stated here in support of this statement. Similarly, in line 218, it states “The ICC higher than 0.90 for most activities and intensities represents excellent reliability”, but the reference for this value of 0.90 is not specified.

18.  The abstract mentions “Index 20 Correlation Coefficient was used to test reliability and Pearson's correlation for validity” but the article states “Interclass Correlation Coefficient”. This is not clear.

Author Response

Dear reviewer, Thank you for your thoughtful and relevant review. I send the review point by point.

1.The number of study participants is thirty-one (31), which is a very low number for attaining adequate statistical power. The authors did not mention any rationale in support of this.

Response:  We recruit 48 eligible pregnant women. Sixteen of them do not accomplish with minimum of 480 minutes/day of accelerometer use and one woman had a premature delivery. Line: 156-157

2.In line 121: It states, “Pregnant women with obesity were categorized according to Institute of Medicine into grade I (BMI ≥30; <35 Kg/m2), grade II (BMI ≥35; <39 Kg/m2) and grade III (BMI ≥ 40 Kg/m2) [28].” However, in the reference article [28], table 2-2 titled “Comparison of Institute of Medicine (IOM) and World Health Organization (WHO) BMI Categories” shows that IOM has only Obese Class I (>29 kg/m2), while WHO has Obese Class I, II and III. In addition, the values of these three classes do not match the “grades” mentioned by the present authors in the manuscript. Thus it is not clear on what basis the authors made the referenced statement.

Response:  Pregnant women with obesity were categorized according to World Health Organization into class I (BMI 30-34,9 Kg/m2), class II (BMI between 35-34.9 Kg/m2) and class III (BMI ≥ 40 Kg/m2) [20]. Line: 133-135

3.The quality/resolution of Fig. 1 is too low to be properly understood. Also, the y-axis labels are not properly typed and are occluded by the axes titles.

Response: Solved

4.Table 3 states three levels of PA but Table 4 shows four levels of activities. This is not clear in the text.

Response:  On PPAQ, the intensity below 3 MET’s is subdivided in light and sedentary. To correlate the PPAQ below 3 MET’s intensity with light intensity measured by accelerometry (also <3.0 MET's), we used the sum of light and sedentary PPAQ intensities. Line:149-152

5.In line 225: The authors mentioned “The mean values for the different types of activity are similar to those found in Portuguese validation, but slightly higher compared to the French study, mainly for total and occupational activity, the fact that only 41% of the participants of the French study were employed in the last trimester may have contributed to low energy values spent on occupational activity.” However, there is no reference stated and hence it is not clear which study has been referred to here.

Response:  The mean values for the different types of activity are similar to those found in Portuguese PPAQ validation [37], but slightly higher compared to the Chandonnet et al., (2012) study, which aimed measure PA of pregnant women with obesity, mainly for total and occupational activity, the fact that only 41% of the participants in Chandonnet et al., (2012) study were employed in the last trimester may have contributed to low energy values spent on occupational activity. It was also found that most of the activities of pregnant women with obesity are domestic and occupational, this data are corroborated by other authors [11,38]. Lines 241-248

  1. The participants in the study used the accelerometer for 505 minutes per day (~ 8 h). So, there is no data about the activity of the remaining 16 h of the day, which is a severe limitation of the study. This might also have confounded the findings and interpretation of the results in the present study.

Response:  This article is the result of a larger investigation where week and weekend time of accelerometer use were counted, so the 505 minutes were mean time of use on the weekend and it was a transcription error.  “Pregnant women walked an average of 5479.7 steps per day, and the accelerometer was used on average 678.9 minutes per day (Table 3).” Line 187-189

7.In line 253: The authors state “We can consider the questionnaire itself as a limitation to the present study, since the activities are classified through compendium and these values were obtained in a laboratory with adults (non-pregnant) [18].” This is another severe limitation of the study and makes the basic foundation of the study questionable. In other words, if the questionnaire itself is not accurate, how can the whole study be based on this?

Response: The correlation between the PPAQ and accelerometry was moderate (r=0.435) for moderate intensity, however the values for total activity are considered weak (r=0.271), similar results were found in the validation of the original PPAQ [19]. One potential explanation for the weak values of correlation, aside from recall bias of questionnaire because PPAQ data refer to the last three months, may be the fact that the cut-off points for moderate and vigorous activity being based on non-pregnant and non-obese samples, and different cut-offs may be needed to adequately assess activity during pregnancy given the changes they are experiencing across gestation [19,45]. Line 270-277

  1. How is the domestic and occupational activity been classified and measured? Also, as pointed out above, the accelerometer was used only for ~ 8 h a day with no data available for the remaining 16 h of the day.

Response: Domestic and occupational physical activity was only subjectively evaluated through the PPAQ, the use of accelerometry was approximately 11 hours.

9.The manuscript contains plenty of spelling mistakes, typographical and grammatical errors, incorrect sentence constructions, and incorrecter use of punctuations. It also contains sudden and abrupt changes of paragraphs, and in many places only one sentence is written in a paragraph format. Overall, it is very poorly written and needs extensive corrections. [For reference, one such example of an incorrect sentence construction is “After seven days of use, the accelerometers were collected and downloaded the data.”]

Response:  Reviewed

10.In line 83: It states, “international PA recommendations”. The name of the article or the source is not mentioned. Assuming it is the ACSM, it needs to be specified. In addition, ACSM has not been spelled out at the first place in the article.

Response:  In the absence of obstetric or medical complications, the exercise recommendations during pregnancy are consistent with recommendations for healthy adults [22]. The  American College of Sports Medicine, the Canadian guidelines for PA throughout pregnancy and the American College of Obstetricians and Gynecologists, recommend that pregnant women, including pregnant women with overweight and obesity, should accumulate at least 150 minutes of moderate-intensity PA each week to achieve clinically meaningful health benefits and reduce pregnancy complications [22–24].  Line 89-95

11.The article also did not explain clearly about the energy expenditure calculation.

Response:  The energy expenditure on the activity in MET's (intensity) is multiplied by duration of activity per day and thus obtains average measurement of energy spent weekly (MET-h•wk-1) [25]. Line:101-103

To interpret the accelerometer data we used Matthews’s cut-point protocol [30], (<759 counts/min- Light intensity (<3.0 MET's); between 760- 5725 counts/min- Moderate intensity (≥ 3.0 and <6.0 MET's); ≥ 5726 counts/min- Vigorous intensity (≥ 6.0 MET's)), line:124-126

12.The results are not written specifically. For example, the authors mentioned at one place “It was also found that most of the participants were primiparous.” without specifying the value. In other words, it is not clarified what is meant by “most” in the result, however, table 1 states it is 64.5%.

Response:  Table 1 shows the sociodemographic characteristics of the sample and obstetric history. Sociodemographic analysis found that a mean of pre-pregnancy BMI was 36.52kg/m2 and most of the pregnant women (67.7%) had an BMI equal or higher than 35kg/m2.

The mean of gestational age of pregnant women was 21.5 weeks, 13 (41.9%) pregnant women were in the second trimester and 10 (32.3/) on third trimester . It was also found that 64.5% of the participants were primiparous. Line: 159-164

13.The results did not specify gravida (i.e. the number of pregnancies), but only the parity (i.e. the number of live birth) of the pregnant participants.

Response: Data added to table1

14.The table 1 shows “ = “ instead of “ ) “ in the Parity >= 1 row.

Response: Reviewed

15.In line 154: It states, “Moderate activities correspond to 39.9% of the total activities, the vigorous and sports activities were the ones with the lowest values, 0.4% (1.29 MET-h•wk-1) and 2.2% (6.43 MET-h•wk-1) respectively”. However, it is confusing why the authors mentioned “lowest” when two values (0.4% and 2.2%) are indicated.

Response:  Moderate intensity activities correspond to 39.8% of the total activities and the vigorous intensity activities representes only 0.4% (1.29 MET-h•wk-1) of energy expedure, regarding the type of activity, Sport/Exercise represent 2.2% (6.43 MET-h•wk-1) of the energy expendure. Domestic activity and moderate intensity had a positively skewed, and we found outliers especially in vigorous intensity (Figure 1). Lines:172-176

16.In Table 3: It shows the accelerometer use was 505.1 min per day, but the total activity was 678.1 min per day. It is not clear how the total activity was measured when the accelerometer was not used during the day.

Response:  Reviewed

17.In line 193: It states, “The vigorous intensity score was the one with a lower Intraclass Correlation Coefficient of 0.58 considered moderate reliability”. Apart from the fact that the sentence is not constructed correctly, it is not clear how the value of 0.58 was considered. There is no reference stated here in support of this statement. Similarly, in line 218, it states “The ICC higher than 0.90 for most activities and intensities represents excellent reliability”, but the reference for this value of 0.90 is not specified.

Response:  The Intraclass Correlation Coefficient and the Standard Measurement Error were used to test the reliability of the PPAQ. The values <0.5 as "poor", ≥0.5 and <0.75 as "moderate", ≥0.75 and <0.9 as "good" and ≥0.90 with "excellent" [33] Line:144-146

18.The abstract mentions “Index 20 Correlation Coefficient was used to test reliability and Pearson's correlation for validity” but the article states “Interclass Correlation Coefficient”. This is not clear.

Response:  Transcription error: Reviewed

Reviewer 2 Report

Dear authors,

The aim of this study is to evaluate the reliability and validity of the Portuguese version of the Questionnaire on Physical Activity and Pregnancy (PPAQ) in pregnant women with obesity and to classify the PA of pregnant women with obesity through PPAQ questionnaire and accelerometry.

It is a robust research with an interesting research topic, however some methodological and drafting shortcomings should be minor corrected.

Please, see the minor revision in attachment.

Good work!

Author Response

Dear reviewer, Thank you for your thoughtful and relevant review. I send the review point by point.

  1. Lines 2 5 (Title): Why not The relationship between Questionnaire on Physical

Activity and Pregnancy (PPAQ) and Accelerometry ?

Response:  The Relationship between Pregnancy Physical Activity Questionnaire (PPAQ) and  Accelerometry to Classify Physical Activity among Pregnant Women with obesity

  1. Lines 17 18 (Abstract): Please, further specify the baseline characteristics of the

sample (age, data collection, gestational age, hospital or clinic).

Response:  An analytical observational study was carried out between May and August of 2019 in University Hospital Center of São João, with a sample of 31 pregnant women with obesity (30.9 ±4.6 yrs; 36.5 ±4.6kg/m2 of BMI and 21.5 ±9.0 gestational weeks). Lines: 19-22

  1. Lines 52 56 (Introduction): Please, add reference to support assumptions ( e.g.,

there are scarcity of PA measurement instruments in pregnant women with

obesity ).

Response: Line 60

  1. Lines 88 90 (Material and methods): Are the recommendations/guidelines for

physical activity and sedentary behavior the same for pregnant women as for the

general population? Did the authors take these differences into account? This may

be a critical point for the extent of the results (Please, see:

https://www.sciencedirect.com/science/article/abs/pii/S170121631830567X).

Response: In the absence of obstetric or medical complications, the exercise recommendations during pregnancy are consistent with recommendations for healthy adults [22]. The  American College of Sports Medicine, the Canadian guidelines for PA throughout pregnancy and the American College of Obstetricians and Gynecologists, recommend that pregnant women, including pregnant women with overweight and obesity, should accumulate at least 150 minutes of moderate-intensity PA each week to achieve clinically meaningful health benefits and reduce pregnancy complications [22–24].  Line 89-95

  1. Lines 234-247 (Discussion): Please, establish some relation between the low

levels of physical activity in the studied population and the development of

hypokinetic and cardio-metabolic diseases (Please, see:

https://www.mdpi.com/1660-4601/19/6/3384)

Response:  In general population, a study shows a positive relationship between low PA levels and hypokinetic and cardio-metabolic diseases [43] . Vigorous activities are extremely low in this specific population of pregnant women [44] and in women with obesity in general [45]. Line: 254-257

  1. Lines 268 271 (Conclusion): The questionnaire is reliable for the population of

pregnant women with obesity and moderately valid. What is the final conclusion?

Can the PPAQ be used instead of accelerometry (more expensive and

inaccessible)?

Response: The Portuguese version of PPAQ questionnaire is reliable for the population of pregnant women with obesity and moderately valid For research requiring a detailed assessment of PA, the questionnaire can be use instead of accelerometer once is accurate, is easy to apply, is less expensive and more accessible. Line: 288-291

  1. References and style/structure References do not comply with IJERPH/MDPI

standards (e.g, line 98). Also, a reference update should be considered.

Response: Reviewed

Reviewer 3 Report

Thank you very much for the opportunity to review the study.

1 The abstract is structured incorrectly. Please elaborate on all abbreviations. Please do not describe the statistical methods used and only describe the results and key findings.
INTRODUCTION
2. "dramatic" is not a very scientific term please replace it with another one.
MATERIAL AND METHODS
3. please divide this section over subsections e.g. participants, study design, statistical analysis.
RESULTS
4. figure 1 is very illegible. Please improve the quality.
DISCUSSION
5. please very much compare the results to more recent global studies.
CONCLUSIONS
6.The conclusions are very truncated, please describe the practical results from the study.
REFERENCES
7 The format of the transcript is incorrect. No recent items which significantly affects the value of the study. No items after 2020.

Author Response

Dear reviewer, Thank you for your thoughtful and relevant review. I send the review point by point.

1.The abstract is structured incorrectly. Please elaborate on all abbreviations. Please do not describe the statistical methods used and only describe the results and key findings.

Response:  The PA of the participants was evaluated through accelerometer and PPAQ questionnaire in two moments (Visit 1: moment of consultation and Visit 2: seven days after, with accelerometer- retest), interclass correlation coefficient was used to test reliability between the PPAQ filled at visit1 and PPAQ filled at visit2, and Pearson's correlation for validity between PPAQ and accelerometry. The interclass correlation coefficient values for total activity were 0.95, 0.97 for moderate and 0.58 for vigorous intensities. It ranged from 0.74 for sports/exercise to 0.96 for domestic activities. Pearson’s correlations shown that PPAQ is moderately valid for moderate intensity (r=0.435). Sixty-seven-point-seven percent of the pregnant women comply with international PA recommendations. Line 22-30

INTRODUCTION
2. "dramatic" is not a very scientific term please replace it with another one.

Response:  In recent years, industrialized nations have experienced an exponential increase of obesity. Line 34

,MATERIAL AND METHODS
3. please divide this section over subsections e.g. participants, study design, statistical analysis.

Response: Reviewed

RESULTS
4. figure 1 is very illegible. Please improve the quality.

Response:  Reviewed

DISCUSSION
5. please very much compare the results to more recent global studies.

Response: Reviewed

CONCLUSIONS
6.The conclusions are very truncated, please describe the practical results from the study.

Response:  The Portuguese version of PPAQ questionnaire is reliable for the population of pregnant women with obesity and moderately valid For research requiring a detailed assessment of PA, the questionnaire can be use instead of accelerometer once is accurate, is easy to apply, is less expensive and more accessible. Line: 288-291

REFERENCES
7 The format of the transcript is incorrect. No recent items which significantly affects the value of the study. No items after 2020.

Response: Reviewed

Round 2

Reviewer 1 Report

The authors provided a response to my earlier comments but unfortunately have not been able to address my concerns adequately and satisfactorily.

Author Response

Dear reviewer, Thank you for your thoughtful and relevant review,

We did a general review of our paper and introduced new bibliographic references, restructured the methods and improved the conclusions.
We send the document for further consideration.
Thank you